# When Does Pairs Trading Outperform Cross-Sectional Momentum?

**Oh Kang Kwon** [1,*] and **Stephen Satchell** [1,2]

1   Discipline of Finance, Codrington Building (H69), The University of Sydney, Sydney, NSW 2006, Australia
2   Trinity College, University of Cambridge, Cambridge CB2 1TQ, UK
*   Correspondence: ohkang.kwon@sydney.edu.au

**Abstract:** In this paper, we analyze the relative performances of pairs trading and cross-sectional momentum (CSM) strategies by comparing their expected returns. It is shown that the Sharpe ratio and the autocorrelation in the spread between the asset returns are the key factors in determining the relative performances of the two strategies, and an analytic expression for the condition under which one strategy outperforms the other is obtained in terms of these factors. It is also shown that the pairs trading strategy outperforms the CSM strategy in the majority of practically relevant situations.

**Keywords:** cross-sectional momentum; pairs trading; Sharpe ratio; spread autocorrelation

## 1. Introduction

Pairs trading and cross-sectional momentum (CSM) strategies are popular investment strategies that rely, in some sense, on opposite assumptions on the behavior of asset returns. The former assumes that when the prices of two closely related assets diverge, they will eventually revert. This is equivalent to the assumption that the asset that overperforms will underperform over a subsequent period. The latter, in contrast, assumes that an asset that overperforms will continue to overperforms. These strategies are very widely used by practitioners, especially in the parts of the market where active management is important. Both, in their simplest forms, are long-short strategies with net zero investment and are likely to be used, for example, by hedge funds.

Although there is a large volume of empirical research into the properties of both strategies, theoretical research investigating the relative performances of these strategies has not been attempted to the best of our knowledge. Refer, for example, to Elliott et al. (2005), Gatev et al. (2003), Grauer (2008), Do and Faff (2010), Zhu et al. (2021) for further details on pairs trading; and Lo and MacKinlay (1990), Jegadeesh and Titman (1993, 2001), Lewellen (2002), Moskowitz et al. (2012), Israel and Moskowitz (2013), and Kwon and Satchell (2020) for further details on CSM.

In this paper, we provide a theoretical comparison of the expected returns on the two strategies by identifying the key factors and deriving an analytic expression for the condition under which one outperforms the other. To do this, we assume that asset returns are jointly normally distributed, which we acknowledge is somewhat restrictive but necessary to derive simple analytic expressions. The Sharpe ratio and the autocorrelation of the spread in the underlying asset returns emerge as the key factors, and the condition is expressed in terms of these quantities. It is also shown that this condition is highly sensitive to the probability of the asset prices reverting, so that even a small change in the probability results in a significant change to the relative performances of the two strategies. Despite this, it is established that in the majority of the practically relevant situations where the expected spread is positive and the spread autocorrelation is small, the pairs trading strategy outperforms the CSM strategy.

The remainder of this paper is organized as follows: Section 2 compares the CSM strategy with the perfect pairs trading strategy where the asset prices revert with certainty. Section 3 extends the analysis to the imperfect pairs trading case under which the asset prices may not revert over a subsequent period, and the paper concludes with Section 4.

## 2. Comparison of Perfect Pairs Trading and CSM

Let $r_t \in \mathbb{R}^2$ be the 2-dimensional vector of asset returns over the period $t$, and assume that the returns are normally distributed and stationary, so that $(r_t, r_{t+1})$ is a 4-dimensional vector and

$$\begin{bmatrix} r_t \\ r_{t+1} \end{bmatrix} \sim \mathcal{N}\left( \begin{bmatrix} \mu \\ \mu \end{bmatrix}, \begin{bmatrix} \Sigma & \Lambda \\ \Lambda' & \Sigma \end{bmatrix} \right), \tag{1}$$

where $\mathcal{N}(m, \Omega)$ denotes a multivariate normal distribution with mean $m$ and covariance matrix $\Omega$,

$$\mu = \begin{bmatrix} \mu_1 \\ \mu_2 \end{bmatrix} \in \mathbb{R}^2, \quad \Sigma = \begin{bmatrix} \sigma_1^2 & \rho\sigma_1\sigma_2 \\ \rho\sigma_1\sigma_2 & \sigma_2^2 \end{bmatrix} \in \mathbb{R}^{2\times2}, \quad \Lambda = \begin{bmatrix} \varsigma_{1,1} & \varsigma_{1,2} \\ \varsigma_{2,1} & \varsigma_{2,2} \end{bmatrix} \in \mathbb{R}^{2\times2}, \tag{2}$$

and $|\rho| \leq 1$. If we denote by $\eta_t$ the spread between the two assets returns, so that

$$\eta_t = r_{t,1} - r_{t,2}, \tag{3}$$

where $r_t = (r_{t,1}, r_{t,2})$, and define $\mu_\eta = \mu_1 - \mu_2$ and $\sigma_\eta^2 = \sigma_1^2 + \sigma_2^2 - 2\rho\sigma_1\sigma_2$, then it follows from (1) that the mean, $\mathbb{E}[\eta_t]$, and variance, $\mathbb{V}[\eta_t]$, of $\eta_t$ are given by

$$\mathbb{E}[\eta_t] = \mu_1 - \mu_2 = \mu_\eta, \tag{4}$$

$$\mathbb{V}[\eta_t] = \sigma_1^2 + \sigma_2^2 - 2\rho\sigma_1\sigma_2 = \sigma_\eta^2. \tag{5}$$

Moreover, if we denote by $\varrho$ the auto-correlation of $\eta_t$, then we have

$$\varrho = \frac{\varsigma_{1,1} + \varsigma_{2,2} - \varsigma_{1,2} - \varsigma_{2,1}}{\sigma_\eta^2}. \tag{6}$$

Finally, if we denote by $\mu_{\text{CSM},t+1}$ the expected return from a 2-asset cross-sectional (CSM) momentum strategy, then it follows from Kwon and Satchell (2018) Equation (13) that

$$\mu_{\text{CSM},t+1} = \mu_\eta \left( 2\Phi\left(\frac{\mu_\eta}{\sigma_\eta}\right) - 1 \right) + 2\varrho\sigma_\eta\phi\left(\frac{\mu_\eta}{\sigma_\eta}\right), \tag{7}$$

where $\phi$ and $\Phi$ denote the standard normal probability density and cumulative distribution functions, respectively, and we note that the fundamental quantities that determine the expected returns from the pairs trading and the CSM strategies were $\mu_\eta, \sigma_\eta$, and $\varrho$. It follows from (4) and (5) that $\mu_\eta/\sigma_\eta = (\mu_1 - \mu_2)/\sigma_\eta$ is effectively the Sharpe ratio corresponding to the portfolio consisting of a long position in the first asset and a short position in the second asset, and given the popularity of the Sharpe ratio with practitioners, we define

$$\gamma_\eta = \frac{\mu_\eta}{\sigma_\eta}, \tag{8}$$

and rewrite $\mu_{\text{CSM},t+1}$ in terms of $\gamma_\eta$ so that

$$\mu_{\text{CSM},t+1} = \gamma_\eta\sigma_\eta\left(2\Phi(\gamma_\eta) - 1\right) + 2\varrho\sigma_\eta\phi(\gamma_\eta). \tag{9}$$

It follows immediately from (9) that if $\gamma_\eta > 0$ and $\varrho > 0$, then $\mu_{\text{CSM},t+1} > 0$. That is, if the first asset has a higher expected return and auto-correlation of the spread, $\eta_t = r_{t,1} - r_{t,2}$, is positive, then the expected return from the CSM strategy is positive.

We now examine the sensitivity of the expected CSM return with respect to the parameters $\varrho$ and $\gamma_\eta$. Firstly, we have

$$\frac{\partial \mu_{\mathrm{CSM},t+1}}{\partial \varrho} = 2\sigma_\eta \phi(\gamma_\eta) > 0,$$

so that the expected CSM return is an increasing function of the spread auto-correlation $\varrho$. Next, we have

$$\frac{\partial \mu_{\mathrm{CSM},t+1}}{\partial \gamma_\eta} = \sigma_\eta\big(2\Phi(\gamma_\eta) - 1\big) + 2\gamma_\eta \sigma_\eta \phi(\gamma_\eta) - 2\varrho\sigma_\eta \gamma_\eta \phi(\gamma_\eta)$$

$$= 2\gamma_\eta \sigma_\eta \phi(\gamma_\eta)(1 - \varrho) + \sigma_\eta\big(2\Phi(\gamma_\eta) - 1\big),$$

so that CSM return is increasing in $\gamma_\eta$ if $\gamma_\eta > 0$ and decreasing in $\gamma_\eta$ if $\gamma_\eta < 0$. In particular, it then follows that for each fixed $\varrho$ and $\sigma_\eta$, the minimum of $\mu_{\mathrm{CSM},t+1}$ occurs at $\gamma_\eta = 0$, or equivalently when $\mu_1 = \mu_2$, with corresponding value $\varrho\sigma_\eta\sqrt{2/\pi}$.

If the investor knew with certainty that $\mu_1 > \mu_2$, so that $\gamma_\eta > 0$, then the investor could construct a portfolio at time $t$ consisting of a long position in the first asset and a short position in the second asset. Such an investor could be considered to be a perfect-pairs trader, and the return from the strategy at time $t + 1$ would be $\eta_{t+1} = r_{t+1,1} - r_{t+1,2}$, which is normal with mean $\mu_\eta = \mu_1 - \mu_2$ and variance $\sigma_\eta^2$. From (8) and (9), the difference between the expected return on the pairs trading strategy and the expected CSM return is

$$\mu_\eta - \mu_{\mathrm{CSM},t+1} = 2\gamma_\eta \sigma_\eta\big(1 - \Phi(\gamma_\eta)\big) - 2\varrho\sigma_\eta \phi(\gamma_\eta)$$

$$= 2\sigma_\eta \phi(\gamma_\eta)\left(\frac{\gamma_\eta\big(1 - \Phi(\gamma_\eta)\big)}{\phi(\gamma_\eta)} - \varrho\right), \tag{10}$$

and this difference is clearly positive if $\varrho \le 0$. This condition is intuitively clear, since pairs trading relies on asset prices that diverged over a prior period to revert back to their common mean, whereas the CSM strategy relies on the opposite being the case. We now proceed to analyze the difference (10) in more detail, and begin by recognizing that the first term inside the parentheses in (10) is related to the so-called Mill's ratio, $M(\gamma_\eta)$, defined by

$$M(\gamma_\eta) = \frac{1 - \Phi(\gamma_\eta)}{\phi(\gamma_\eta)}. \tag{11}$$

It follows from a result on page 132 of Sampford (1953) that if $\gamma_\eta > -1$, then

$$M(\gamma_\eta) < \frac{4}{3\gamma_\eta + \sqrt{8 + \gamma_\eta^2}} = U(\gamma_\eta), \tag{12}$$

and since it is assumed that $\gamma_\eta > 0$, Mill's ratio will satisfy (12) in the case of a perfect-pairs trader. In what follows, we make use of the upper bound, $U(\gamma_\eta)$, in (12) to obtain a condition under which CSM outperforms the perfect-pairs trading, and so it is of interest to examine how close $\gamma_\eta U(\gamma_\eta)$ is to $\gamma_\eta M(\gamma_\eta)$. The plot of the two functions in Figure 1 shows that $\gamma_\eta U(\gamma_\eta)$ is indeed a tight upper bound for $\gamma_\eta M(\gamma_\eta)$, with the maximum difference between $\gamma_\eta U(\gamma_\eta)$ and $\gamma_\eta M(\gamma_\eta)$ in the region $\gamma_\eta \ge 0$ being approximately 0.02 at $\gamma_\eta = 0.36$.

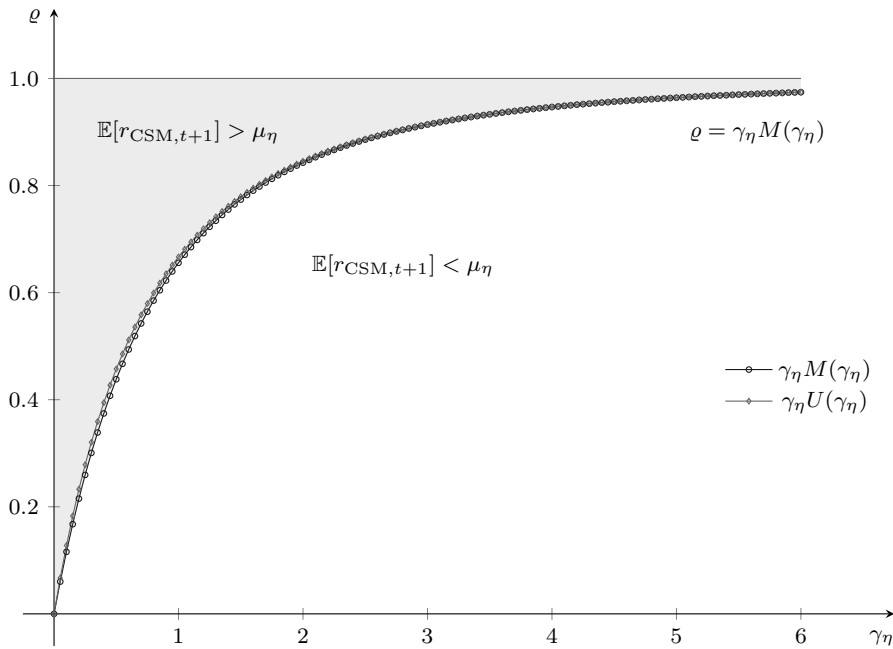

**Figure 1.** Plot of $\gamma_\eta M(\gamma_\eta)$ and $\gamma_\eta U(\gamma_\eta)$ as functions of $\gamma_\eta$, where $U(\gamma_\eta)$ is the upper bound of $M(\gamma_\eta)$ in (12).

Now, we have from (10) that

$$\mu_\eta - \mu_{\text{CSM},t+1} = 2\sigma_\eta \phi(\gamma_\eta)\big(\gamma_\eta M(\gamma_\eta) - \varrho\big) < 2\sigma_\eta \phi(\gamma_\eta)\left(\frac{4\gamma_\eta}{3\gamma_\eta + \sqrt{8 + \gamma_\eta^2}} - \varrho\right),$$

and since $\sigma_\eta \phi(\gamma_\eta) > 0$, a sufficient condition for $\mu_\eta < \mu_{\text{CSM},t+1}$ is

$$\frac{4\gamma_\eta}{3\gamma_\eta + \sqrt{8 + \gamma_\eta^2}} < \varrho. \tag{13}$$

Given that for a perfect-pairs trader $\gamma_\eta \geq 0$, this is equivalent to

$$4\gamma_\eta < \varrho\left(3\gamma_\eta + \sqrt{8 + \gamma_\eta^2}\right) \quad \Leftrightarrow \quad 0 < \left(1 - \gamma_\eta^2\right)\varrho^2 + 3\gamma_\eta^2\varrho - 2\gamma_\eta^2.$$

Denoting by $q(\varrho)$ the quadratic

$$q(\varrho) = \left(1 - \gamma_\eta^2\right)\varrho^2 + 3\gamma_\eta^2\varrho - 2\gamma_\eta^2, \tag{14}$$

we have that (13) is equivalent to $q(\varrho) > 0$. In order to proceed further, we must consider three cases, viz., $\gamma_\eta < 1$, $\gamma_\eta = 1$, and $\gamma_\eta > 1$. Firstly, if $\gamma_\eta = 1$, then $q(\varrho) = \gamma_\eta^2(3\varrho - 2)$, and (13) reduces to $3\varrho - 2 > 0$ so that the condition becomes

$$\frac{2}{3} < \varrho \leq 1. \tag{15}$$

Next, note that if $\gamma_\eta \neq 1$, then the roots, $\varrho_\pm(\gamma_\eta)$, of $q(\varrho)$ are real and given by

$$\varrho_\pm(\gamma_\eta) = \frac{-3\gamma_\eta^2 \pm \gamma_\eta\sqrt{8 + \gamma_\eta^2}}{2(1 - \gamma_\eta^2)}. \tag{16}$$

If $\gamma_\eta < 1$, then $q(\varrho)$ is concave, and taking into account the restriction $|\varrho| \leq 1$, we have $q(\varrho) > 0$ if and only if $\varrho > \varrho_+(\gamma_\eta)$ so that the condition on $\varrho$ in this case is

$$\varrho_+(\gamma_\eta) = \frac{-3\gamma_\eta^2 + \gamma_\eta\sqrt{8 + \gamma_\eta^2}}{2(1 - \gamma_\eta^2)} < \varrho \leq 1, \tag{17}$$

where $\varrho_+(\gamma_\eta)$ is positive since $0 \leq \gamma_\eta < 1$. Finally, if $\gamma_\eta > 1$, then $q(\gamma_\eta)$ is convex, but surprisingly the condition on $\varrho$ is the same as for the case $\gamma_\eta < 1$. In summary, if we define $\varrho_+(1) = \frac{2}{3}$, where $\varrho_+(\gamma_\eta)$ is as given in (16), then $\mu_{\text{CSM},t+1} > \mu_\eta$ if $\varrho_+(\gamma_\eta) < \varrho \leq 1$.

As shown in Figure 1, the interval $(\varrho_+(\gamma_\eta), 1]$ decreases with $\gamma_\eta$ so that the range of values of $\varrho$ over which the perfect pairs trading strategy outperforms the CSM strategy increases with $\gamma_\eta$. It is worth noting that the Sharpe ratios reported in Gatev et al. (2003), obtained by considering pairs trading strategies using US stocks from 1962 to 2002, lie in the range $0.35 \leq \gamma_\eta \leq 0.59$. For the CSM strategy to outperform the pairs trading strategy over such a range of $\gamma_\eta$, the corresponding $\varrho$ will need to be in the range $0.34 \leq \varrho \leq 0.52$. We note that $\varrho$ is the autocorrelation in the spread and not an individual asset autocorrelation.

## 3. Comparison of Imperfect Pairs Trading and CSM

We now assume that pairs trading is not perfect so that the assumed condition $\mu_\eta = \mu_1 - \mu_2 > 0$ is no longer certain, but instead holds with some probability $p \in [0, 1]$. Since the asset prices that diverged over a prior period do not always revert, this is perhaps the situation that is more likely to be of practical relevance. The expected return, $\mu_\eta(p)$, on the pairs trading strategies in this case, is then

$$\mu_\eta(p) = p(\mu_1 - \mu_2) + (1 - p)(\mu_2 - \mu_1) = (2p - 1)(\mu_1 - \mu_2) = (2p - 1)\mu_\eta(1). \tag{18}$$

Moreover, the difference between the expected returns from the imperfect pairs trading and the CSM strategy is now

$$\mu_\eta(p) - \mu_{\text{CSM},t+1} = 2\sigma_\eta \phi(\gamma_\eta)\left(\frac{p - \Phi(\gamma_\eta)}{\phi(\gamma_\eta)} - \varrho\right), \tag{19}$$

and it follows that $\mu_\eta < \mu_{\text{CSM},t+1}$ if and only if

$$\varrho > \frac{p - \Phi(\gamma_\eta)}{\phi(\gamma_\eta)}. \tag{20}$$

The impact of the probability $p$ on the right-hand side of (20), and hence on $\mu_\eta(p) - \mu_{\text{CSM},t+1}$, is plotted in Figure 2, and we see that the performance of the pairs trading strategy relative to the CSM strategy is extremely sensitive to $p$. In fact, even the slightest uncertainty in the required condition $\mu_\eta > 0$ for the pairs trading strategy to be fully effective results in a significant change to the region in the $(\gamma_\eta, \varrho)$ space over which the pairs trading outperforms CSM.

Before closing this section, we note that the situation under which pairs trading would most likely be employed in practice is where $\mu_\eta > 0$ and are small, and $\varrho < 0$. Since this corresponds to the case where the two assets have similar expected returns, any difference in the returns over a prior period is likely to reverse over the subsequent period, and the reversal will result in $\mu_\eta > 0$. This is the region labeled $\mathcal{R}_-$ in Figure 2, and as expected, pairs trading outperforms the CSM strategy in this region. The situation in which the CSM strategy would be appropriate is where $\mu_\eta > 0$ and $\varrho > 0$, which corresponds to the region labeled $\mathcal{R}_+$ in Figure 2. It is interesting to note that in much of the practically relevant subregion of $\mathcal{R}_+$ where $\varrho$ is small, the pairs trading strategy still outperforms the CSM strategy other than when $\gamma_\eta \approx 0$.

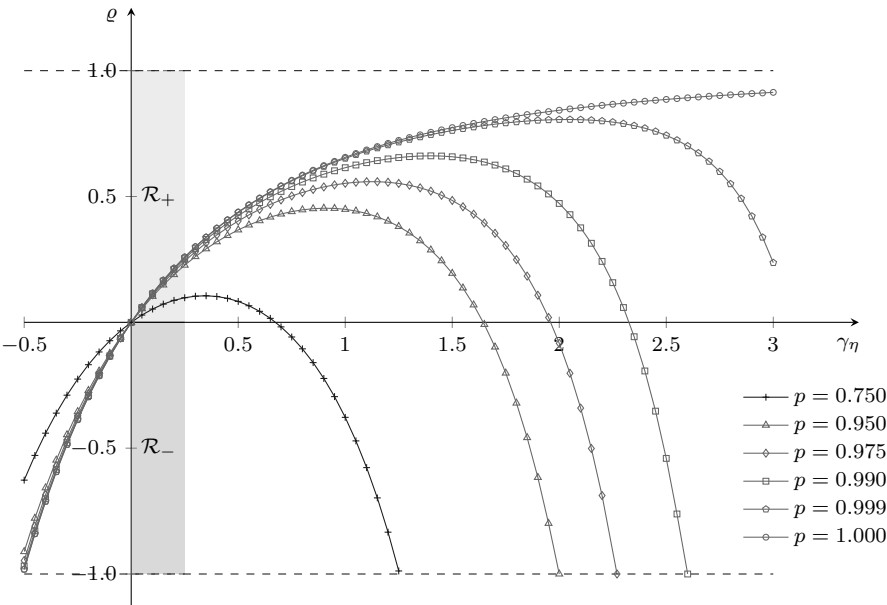

**Figure 2.** Plot of $(p - \Phi(\gamma_\eta))/\phi(\gamma_\eta)$ as a function of $\gamma_\eta$ for various values of $p$.

## 4. Conclusions

In this paper, the relative performances of pairs trading and cross-sectional momentum (CSM) strategies were investigated in terms of their expected returns. The Sharpe ratio, $\gamma_\eta$, and the autocorrelation, $\varrho$, in the asset return spread were identified as the key factors that determine the performances of the two strategies, and an analytic condition specifying the region in the $(\gamma_\eta, \varrho)$ space over which one strategy outperforms the other was derived in terms of these factors.

It was also shown that although the performance of the pairs trading strategy is highly sensitive to the probability of the asset prices reverting, it not only outperforms the CSM strategy in situations where it is most likely to be used, but also does in the majority of the practically relevant situations where the CSM strategy would be most appropriate.

**Author Contributions:** Conceptualization, S.S; methodology, O.K.K. and S.S; draft preparation, O.K.K.; review and editing, O.K.K. and S.S. All authors have read and agreed to the published version of the manuscript.

**Funding:** This research received no external funding.

**Data Availability Statement:** Not applicable.

**Conflicts of Interest:** The authors declare no conflict of interest.

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
