# Peer review of "When Does Pairs Trading Outperform Cross-Sectional Momentum?"

_jrfm, doi:10.3390/jrfm15110512_

Round 1

Reviewer 1 Report

The current manuscript is written and presented with few details in the research steps and results. Some major points are required to improve or clarify.

An assumption on which the demonstration is based is "returns are normally distributed". In fact, this assumption is respected only in a few conditions that must be specified.

There are a lot of notations please specify what means this. Even if the notations used are generally valid in the specialized literature, it would be necessary for all the notations in the paper to be defined/specified.

The literature on which the research is based is limited and mostly old.

There are no references to the journal Journal of Risk and Financial Management. This makes me think that the paper is not necessarily related to the scope of the journal

The paper does not respect the template of the Journal.

Reviewer 2 Report

The paper respects the requirements of an academic research article. The topic is very interesting.

The literature review can be improved by adding relevant references from 2018 onward.

I will let fellow reviewers to appreciate the mathematics part.

The conclusions are supported by the results.

Reviewer 3 Report

Most studies of investment strategies are empirical in nature. Therefore, the theoretical study of the efficiency factors of the investment strategies under consideration is of scientific interest.

The following comments will help improve the readability of the manuscript.

For a better understanding of the context of the article, the main points of investment strategies for trading pairs and CSM should be given in the literature review.

In my opinion, it is necessary to clarify the target audience for which the results of the study of the effectiveness of investment strategies can be useful.

For a better understanding of the formulas, it is necessary to explain the notation, which may have different meanings depending on the subject of the study.

It is useful to justify the connection with the Mill’s ratio.

Round 2

Reviewer 1 Report

The bibliography is missing in revised form.

The works cited are not visible in the content of the work.

The suggestion 

"There are a lot of notations please specify what means this. Even if the notations used are generally valid in the specialized literature, it would be necessary for all the notations in the paper to be defined/specified. "

was not fully respected.
